# Repeatability and genetic advances in early maturing maize hybrid trials conducted under *Striga*-infested and non-infested conditions

**Adamu Masari Abubakar**[1,2], **Idris Ishola Adejumobi**[1*], **Kayode Rapheal Fowobaje**[1], **Ibnou Dieng**[1], **Zakariya Saminu**[2], **Simon Imoro**[1], **Baffour Badu-Apraku**[1]

**1** International Institute of Tropical Agriculture (IITA), Ibadan, Oyo State, Nigeria, **2** Department of Plant Science, Institute for Agricultural Research, Ahmadu Bello University, Zaria, Kaduna State, Nigeria

* i.adejumobi@cgiar.org (I.I.A.)

## Abstract

In sub-Saharan Africa (SSA), maize (*Zea mays* L.) is both a cash crop and an important staple crop. However, *Striga hermonthica* infection constrains its production and productivity. A total of 159 hybrids from 21 international trials were evaluated under *Striga*-infested (STRINF) and *Striga* non-infested (STRNON) conditions at Mokwa and Abuja, Nigeria, from 2010 to 2021. The data were used to (i) determine the genetic enhancements in grain yield and *Striga* adaptive traits and (ii) assess the repeatability of the trials in identification of promising hybrids. Significant annual genetic gains in grain yield of 3.40% and 3.71% with increases of 76.87 and 127.02 kg ha⁻1 yr⁻1 were recorded under STRINF and STRNON conditions, respectively. The genetic gains in grain yield were associated with 3.04 and 2.25% decreases in *Striga* damage ratings at 8 and 10 weeks after planting (WAP), respectively, and 1.27% in the number of emerged *Striga* plants at 10 WAP. The results indicated that ears per plant and flowering dates had the highest consistency in repeatability estimates while the number of emerged *Striga* plants at 8 and 10 WAP recorded the lowest consistency in repeatability estimates. Generally, substantial progress has been achieved with a good level of repeatability estimates for the early maturing maize hybrid trials evaluated under STRINF and STRNON conditions. Those results have demonstrated that the breeding strategies adopted during the 12-year breeding period have been effective, and that the potential of the trials in the exploration of the genetic potential of the hybrids for commercialization in the SSA for food security and poverty alleviation.

## Introduction

Maize plays an important role as a staple food in the diets of millions of people in Africa, especially in ensuring food security. Maize production and productivity are faced by a myriad constraints of which *Striga hermonthica* (purple witchweed), an obligate hemi-parasitic plant is one of the major factors causing substantial loss to grain yield on farmers' fields in

**Data availability statement:** The dataset used to generate the results presented in this study is available in the repository with the information below. Name of Repository: Figshare URL: URL https://figshare.com/articles/dataset/GG_STRINF_NONINF_2010-2021_Masari_xlsx/28302458?file=51994256 Name of Data: GG_STRINF&NONINF_2010-2021_Masari.xlsx. figshare. Dataset DOI: https://doi.org/10.6084/m9.figshare.28302458.v1

**Funding:** This research was supported by the Bill & Melinda Gates Foundation under the Accelerated Genetic Gains (AGG) and Stress Tolerant Maize for Africa (STMA) Projects. The funder was the main donor of the AGG and STMA projects and also the payer of the article processing charge for the manuscript.

**Competing interests:** The authors have declared that no competing interests exist.

sub-Saharan Africa (SSA) [1]. Even though *S. hermonthica* parasitizes the root systems of several cereals, maize is the most preferred among staple cereal crops. Every cropping season, soil-borne *Striga* seeds germinate and attach to maize plant root systems, causing plant toxicity and yield reduction by siphoning water and nutrients from the host maize plant, resulting in the death of the host plant in extreme cases. The outcome of this association is a significant reduction in the obtainable grain yield. *Striga* parasitism is more common in the Savanna of SSA resulting in food insecurity and rural poverty. The lifespan of *Striga* depends on its ability to siphon nutrients and water from host plants to promote its growth and development. *Striga* species have 5–7 days to establish themselves on a suitable host, otherwise, if the stored resource in the seeds are exhausted, the radicles will wither [2] *Striga* is a major threat cereals crops. It causes yield losses of $7 to $10 billion worth and, could result in up to 100% losses in maize production [3–6]

Breeding for *Striga* resistance has proven to be the most cost-effective, feasible, and long-term strategy for control by farmers with limited resources. The scientists of the International Institute of Tropical Maize Improvement Program (IITA-MIP) initiated *Striga* resistance research on maize in 1982 with most of the germplasm originating from the U.S. Corn Belt. Resistant inbreds and hybrids of IITA were first identified in 1983 from field testing at Mokwa in Nigeria [7]. Since then, breeders of the Maize Improvement Program (MIP) have been working steadily to develop maize varieties/hybrids with elevated levels of *Striga* tolerance to support smallholder farmers in the targeted production environments. In this regard, IITA-MIP in collaboration with the National Agricultural Research Scientist (NARS) through deliberate teamwork, has developed and released several high-yielding multiple stress-tolerant (MST) (i.e., resistant to *Striga* and tolerant to drought and low N) hybrids and open-pollinated varieties (OPVs), many of which are suitable for several agro-ecologies in SSA. For the past two decades, the IITA-MIP has been developing and deploying *Striga*-tolerant hybrids through regional trials to NARS in West and Central Africa (WCA) for evaluation and commercialization in the sub-region. The hybrids in the regional trials are selected based on successful performance in at least three successive trial evaluations. The regional trials allow the extensive testing of promising maize hybrids across a set of target environments. It offers the National Maize Programs of WCA the opportunity to select hybrids/OPVs with superior performance for further testing and release directly as a replacement for old and less-performing varieties or as source materials for their breeding programs. In addition to being dispatched to the NARS partners annually, IITA-MIP participates annually in the evaluation of these trials across multiple test environments. The *Striga* trials are also conducted annually at Mokwa and Abuja under *Striga*-infested (STRINF) and non-infested (STRNON) conditions.

Assessment of the breeding program through timely estimation of the genetic gains from selection is important to determine the progress that has been made and the effectiveness of the breeding approach used over the years. To demonstrate gains in grain yield and other agronomic attributes from selection, several studies comparing the performance of maize cultivars developed over time and evaluated in field trials conducted in common sets of environments were documented [8–14]. These studies have been conducted by choosing genotypes representative of the breeding eras or cycles and evaluating them simultaneously in common sets of environments over the years. A few of the previous studies published by Menkir and Meseka [13] indicated a yield gain of 3.2% with a mean increase of 93.7 kg ha$^{-1}$ yr$^{-1}$ under *Striga*-infested environments in a set of 32 late/intermediate maturing maize hybrids developed over three breeding periods. Badu-Apraku *et al.* [15] evaluated 54 hybrids developed during three breeding periods of genetic enhancement. The authors reported annual genetic gains from selection in grain yield of 84.72 kg ha$^{-1}$ yr$^{-1}$ (4.05%) and 61 kg ha$^{-1}$ yr$^{-1}$ (1.56%), under multiple-stress and non-stress environments, respectively. These studies necessitate a

substantial resource investment. This includes time, difficulty in producing, packaging, and dispatching the seeds for the trials, logistical difficulties when reassembling historical genotypes. Additionally, genotypes developed years ago may perform differently in most recent years due to climate change resulting in new genotypes being favored. Furthermore, these studies do not facilitate monitoring of the effectiveness of breeding strategies to allow the exploration of the weaknesses to make the necessary adjustments. However, the effects of different trial management techniques and varying climatic conditions on genetic trends are usually eliminated to provide the most unbiased estimates of genetic gain [16].

In estimating genetic gains from selection over several breeding periods, numerous methodologies have been documented. Crespo-Herrera *et al*. [17] used the factor analytic model approach to estimate genetic gains in wheat. This approach regresses the differences in mean grain yield of the highest-yielding varieties on the mean yield of checks. The authors reported an annual genetic gain of 1.67% relative to the commercial check and 0.53% relative to the local check. Additionally, Menkir *et al*. [18] used the mixed model analysis to conduct genetic gain study in late/intermediate maturity maize. The authors reported yield gain of 86.60 kg ha$^{-1}$ yr$^{-1}$ under *Striga* infestation and 102.44 kg ha$^{-1}$ yr$^{-1}$ under *Striga*-free condition. Breeding trials continuously consider the dynamics of adding new genotypes and eliminating the inferior ones. As a result, many authors [18–21] recommended the use of mixed linear model analysis to reliably estimate genetic gains for unbalanced data from multi-environment trials. Recently, The Consortium of International Agricultural Research Centers (CGIAR) has updated its recommendations, emphasizing the use of historical data sets to analyze genetic gain. The strategy is crucial to enable regular monitoring of the progress of a breeding program and the efficacy of breeding methodologies employed for the development and identification of superior genotypes.

The repeatability of the trial observed over the years may differ from that of other years and/or all years combined. The observed discrepancy could result from differences in climate conditions, trial management, or random chance due to the additive effect of the experimental errors. Timely assessment of the efficiency of the methodologies employed in a breeding program through the estimation of the repeatability and the estimates of the realized genetic gains over time in a breeding program is important. It allows the evaluation of the progress that has been made, the approach, and the effectiveness of breeding methodologies. This allows the planning of new strategies or adopting the most appropriate method in the breeding program. Thus, the objectives of the present study were to; (i) determine the genetic improvement in grain yield and *Striga* adaptive traits and (ii) assess the repeatability of the trials in identification of the promising hybrids under both STRINF and STRNON conditions.

## Materials and methods

### Development of the genetic materials

The early and extra-early maize breeding program of IITA started the inbred development in 1994. The program aimed at the development of *Striga* resistant inbred lines, hybrids, and OPVs. The early maturing populations TZE-W Pop DT STR C0, TZE-Y Pop DT STR C0, TZE Comp 5-Y C6, and TZE-W Pop × 1368 STR were initially used as the sources for *Striga* resistance and drought tolerance. These populations were valuable sources of genes for *Striga* resistance and drought tolerance and have been used to develop several outstanding inbreds, hybrids, and OPVs in the early and extra-early breeding program of IITA. However, the levels of resistance to *Striga* in these populations were not as high as desired. In 2007, a program was initiated to increase the frequency of favorable alleles for *Striga* resistance and drought tolerance by using the S1 family recurrent selection breeding scheme to introgressed genes

from the wild maize relative *Zea diploperennis*. This resulted in the development of new generations of outstanding populations that combined improved levels of resistance to *Striga* and tolerance to drought and low soil nitrogen. The approaches that have been utilized in the development of the hybrids used in the present study have been described in detail by [22]. In summary, the germplasm sources were used to generate the S1 lines which were advanced to the S4 stages of inbreeding through repeated selfing and selection. The lines were evaluated under artificial *Striga* infestation and induced moisture stress after each inbreeding cycle. At the S4 stage, 250–300 selected lines were crossed to broad-based testers to estimate the general combining abilities of the lines. Based on the performance in the test-crosses, 90–100 S4 lines were advanced to the S6 stage of inbreeding. The S6 lines were used in development of the hybrids. The sets of hybrids developed, were first evaluated in progeny trials from where successful candidates were advanced to the second testing stage of the progeny trial and then moved to preliminary trials. Based on the genotypic performance, successful candidates were advanced to the regional trials and also evaluated and the data were used for this study.

A total of 159, early maturing white MST hybrids (MST1–159) developed by IITA-MIP, 5 farmer-preferred hybrids marketed conventional *Striga*-resistant hybrids/cultivars (LOC 1–5), and 13 hybrids (SCO 1–13) contributed by multi-national seed companies and commercialized in many African countries were used in the present study. The detailed descriptions and the year of first testing of the hybrids in the regional trial are presented in Supplementary Table S1.

## Management of field trials

In the present study, we used historical data from trials conducted under STRINF and STRNON conditions at Abuja and Mokwa in Nigeria, from 2010 to 2021 growing seasons. The characteristics of the two test environments were presented in Supplementary Table S2. The alpha lattice design was employed for the trials containing more than twenty entries while RCBD was adopted for the trials that had not more than twenty entries. Each trial consisted of three replications with a plot comprising 2 rows, each 4 m long. The spacing between rows was 75 cm while plants within a row were 4 cm apart. Three seeds were placed into each hole and later thinned to two plants per hill resulting in a population density of 66,666 plants ha$^{-1}$. The hybrids were planted in two *Striga*-infested rows and two non-infested rows within each strip, planted exactly opposite each other and separated by a 1.5-meter alley. The artificial *Striga* infestation was achieved by putting about 5000 germinable *Striga* seeds per hill followed by planting maize seeds into the same holes as described in detail by Badu-Apraku and Fakorede [22]. To subject the maize plants to stress and encourage the production of strigolactones that promote good germination of *Striga* seeds and attachment of *Striga* plants to the roots of maize plants in *Striga*-infested plots, fertilizer application was delayed till about 30 days after planting [23]. The compound fertilizer NPK 15:15:15 was applied at the rate of 30 kg ha$^{-1}$ N, P, and K at 30 and 45 days after planting (DAP). Weeds other than *Striga* were removed mannually throughout the growing seasons.

## Traits measured

Data were collected on 10 *Striga*-adaptive traits including grain yield (YLD), days to 50% anthesis (DA), days to 50% silking (DS) anthesis-silking interval (ASI), ears per plant (EPP), ear aspect (EASP), emerged *Striga* plant at 8 WAP (ESP8), emerged *Striga* plant at 10 WAP (ESP10), *Striga* damage ratings at 8 WAP (SDR8) and *Striga* damage ratings at 10 WAP (SDR10). The detail descriptions of the traits studied were presented in Table 1.

**Table 1. Descriptions of the measured traits of early white maize hybrids evaluated in the regional trials under STRINF and STRNON conditions at Abuja and Mokwa, 2010–2021.**

| Trait | Stage | Unit | Description |
|---|---|---|---|
| Grain yield (YLD) | Harvest | Kg/ha | Computed from the weight of shelled grain adjusted to 80% shelling percentage and corrected for 15% moisture content |
| Days to 50% anthesis (DA) | Flowering | Days | Days from planting to 50% pollen shed |
| Days to 50% silking (DS) | Flowering | Days | Days from planting to 50% silk emergence |
| Anthesis-silking interval (ASI) | Flowering | Days | The time interval between 50% anthesis and silking |
| Ears per plant (EPP) | Post-flowering | Numeric | Calculated by dividing the number of harvested ears by the number of plants harvested per plot |
| Ear aspect (EASP) | Harvest | Scale | Scored on a scale of 1–9, with 1 denoting ears that were clean, tidy, uniform, big, and full, and 9 denoting ears with undesirable characteristics |
| Emerged *Striga* plant at 8 and 10 WAP (ESP8 and ESP10) | Post-flowering | Count | The numbers of Striga plants that were counted at 8 and 10 WAP in the Striga-infested plots |
| *Striga* damage ratings at 8 and 10 WAP (SDR8 and SDR10) | Post-flowering | Scale | Scored on per plot basis on a scale of 1–9 where 1 = no damage, indicating normal plant growth and high resistance, and 9 = complete collapse or death of the maize plant |

## Statistical analyses

The historical data used in the analysis of the present study came from trials conducted under *Striga*-infested (STRINF) and *Striga* non-infested (STRNON) conditions. The analysis followed the following steps:

The first step, involved environment-level analyses with the environment is defined as the combination of trial name, location, year, and condition. We used the linear mixed model with the hybrids considered as random effects to estimate the broad-sense heritability a measure of experimental repeatability [24]. Trials with heritability values less than 0.1 were excluded from the estimated genetic gains. After excluding the environments with heritability estimates less than 0.1, another linear mixed model (equation 1) was fitted for the hybrids treated as a fixed effect to estimate the Best Linear Unbiased Estimates (BLUEs), and weight (inverse of squared standard error) for each hybrid used for the second stage of the analyses as follows:

$$y_{ikj} = \alpha + h_i + \varphi_k + \rho_j + \varepsilon_{ikj} \tag{1}$$

where $y_{ikj}$ represents the yield value of the $i$ th hybrid in the $k$ th block and the $j$ th replication. The overall mean is denoted by α, the random effect of the $a$ th hybrid is represented by $h_i$, $\varphi_k$ is the effect of b$^{th}$ replication, $\rho_j$ is the effect of the kth block, and $\varepsilon_{ikj}$ is the random residual effect attributed to the yield value of the $i$ th hybrid in the $k$ th block and in the $j$ th replication that follows the normal distribution $N\left(0, \sigma_e^2\right)$.

Secondly, we used the model represented by equation 2 [20] to analyse the BLUE values of grain yield and other traits obtained from equation 1 under STRINF and STRNON conditions as proposed by Mackay *et al.* [20]

$$Y_{ijklm} = \mu + \zeta_i + \eta_m + \lambda_{jlm} + \beta_{kjlm} + \left(\zeta\eta\right)_{im} + \xi_{lm} + \varepsilon_{ijklm} \tag{2}$$

Where $Y_{ijklm}$ is the BLUE value of yield of the $i$ th hybrid in the $k$ th block, in the $j$ th replication, in the $l$ th location and $m$ th year. The overall mean is represented by $\mu$. The effect of the $i$ th hybrid is denoted by $\zeta_i$ and the effect of the $m$ th year is represented by $\eta_m$; $\lambda_{jlm}$ represents the effect of the $j$ th replication in the $l$ th location and the $m$ th year. Also, $\beta_{kjlm}$ is the effect of the $k$ th block in $j$ th replication in the $l$ th location and $m$ th year. The interaction effect between the $\left(im\right)$ th hybrid and year is represented by $(\zeta\eta)_{im}$, while $\xi_{lm}$ is the effect of location $l$ within year $m$; $\varepsilon_{ijklm}$ is the residual, attributable to the combined effects of within-trial error

and hybrid x location within-year interaction. Heterogeneity in the residual variances was assumed. Thus, we fitted a distinct residual variance for each condition for every combination of location and year.

Lastly, we used the weighted regression to examine trends (realized genetic gain) in the data from the second stage of the analyses. The BLUEs of the hybrids and their year of origin in each condition were used as dependent and independent variables in the regression analyses respectively. We determined the percentage of each condition's genetic gain that could be attributed to genetic factors. The process involved calculating the ratio of the second stage's weighted regression slope plus the regression's y-intercept, and the product of the slope and year of the initial testing. The p-value < 0.05 for each trait from the summary of the weighted regression model was considered significant. The grain yield performance of the MST, LOC and SCO hybrids were compared using the Boxplot. Similarly, the repeatability estimate of each trait over the years by location and condition was presented using a scatter plot. All analyses were performed in R [25] using the ASReml package [26]

## Results

### Genetic gains in grain yield and *Striga* adaptive traits

During the 12 years of development, evaluation and selection of the early maturing hybrids in IITA-MIP, significant annual gains in grain yield of 3.40 and 3.71% with changes of 76.87 and 127.02 kg ha⁻1 yr⁻1 were recorded under STRINF and STRNON conditions, respectively. The significant increases in grain yield were associated with significant increases of 1.84 and 1.79% in ears per plant and significant decreases of -1.31 and -1.75% in the ear aspect under STRINF and STRNON conditions, respectively. Under the STRINF condition, significant genetic gains of -3.04 and -2.25% in *Striga* damage rating at 8 and 10 WAP, respectively, and -1.27% in the number of emerged *Striga* plants at 10WAP were obtained (Table 2). However, there were no significant gains in anthesis silking interval under STRINF environments and days to anthesis and silking under STRINF and STRNON conditions (Table 2). Additionally, the regression line considering the hybrid's first year of evaluation (independent variable) and traits (dependent variable) showed a positive linear relationship for grain yield and EPP and a negative linear relationship for ESP10, SDR10, and EASP. The breeding period under both conditions did not affect the number of days to flowering and ASI across STRINF and STRNON conditions (Fig 1a–1f).

### Grain yield performance of early maturing maize hybrids evaluated in the regional trials across STRINF and STRNON conditions, 2010–2021

In the present study, three types of hybrids were evaluated, MST, SCO and LOC. The LOC cultivars comprised five hybrids, among them three common cultivars were evaluated over the period of 12 years. Additionally, the MST hybrids were evaluated for 2–6 years making the data well connected to allow an estimate of realized genetic gains. We assessed the genetic improvement under STRINF and STRNON conditions by comparing the MST hybrids' average grain yield performance to the SCO and LOC hybrids (Fig 2). The results showed that, under STRINF, the mean grain yield of MST hybrids ranged from 1091 kg ha⁻1 for hybrid developed in 2012–4287 kg ha⁻1 for hybrid developed in 2018. Similarly, the mean grain yield for SCO hybrids varied from 965 kg ha⁻1 to 2964 kg ha⁻1 and grain yield of LOC hybrids varied from 1914 kg ha⁻1 to 2644 kg ha⁻1 (Supplementary Table S1). On another hand, mean grain yield of the hybrids evaluated under STRNON condition ranged from 2268 kg ha⁻1 to 6861 kg ha⁻1 for the MST hybrids and from 3590 kg ha⁻1 to 6644 kg ha⁻1 for SCO hybrids while 2392 kg ha⁻1 to 4088 kg ha⁻1 were recorded for LOC hybrids. Generally, the MST

**Table 2. Percent genetic gains in grain yield and other *Striga*-adaptive traits of maize hybrids evaluated under STRINF and STRNON conditions in Nigeria between 2010–2021.**

| Trait | Management | % Genetic Gain (year⁻¹) | slope | slope.sd | intercept |
|---|---|---|---|---|---|
| YLD | STRINF | 3.40** | 76.87 | 13.32 | -152175.53 |
|  | STRNON | 3.71** | 127.02 | 21.65 | -252000.76 |
| DA | STRINF | 0.12ns | 0.17 | 0.03 | -283.37 |
|  | STRNON | 0.11ns | 0.06 | 0.04 | -63.30 |
| DS | STRINF | 0.10ns | 0.05 | 0.03 | -52.78 |
|  | STRNON | 0.11ns | 0.06 | 0.05 | -72.47 |
| ASI | STRINF | -0.60ns | -0.01 | 0.02 | 32.56 |
|  | STRNON | -5.62** | -0.13 | 0.02 | 272.06 |
| EASP | STRINF | -1.31** | -0.07 | 0.01 | 145.94 |
|  | STRNON | -1.75** | -0.08 | 0.02 | 167.23 |
| EPP | STRINF | 1.84** | 0.01 | 0.00 | -25.00 |
|  | STRNON | 1.79** | 0.01 | 0.00 | -26.62 |
| ESP8 | STRINF | -0.97ns | -0.31 | 0.22 | 653.83 |
| ESP10 | STRINF | -1.27* | -0.45 | 0.20 | 948.38 |
| SDR8 | STRINF | -3.04** | -0.16 | 0.02 | 328.14 |
| SDR10 | STRINF | -2.25** | -0.13 | 0.02 | 274.04 |

Sd:standard deviation;

**: Significant; ns: Not Significant; YLD: grain yield; DA: days to 50% anthesis; DS: days to 50% silking; ASI: anthesis-silking interval; EPP: ears per plant; EASP: ear aspect; ESP8: emerged *Striga* plant at 8 WAP; ESP10: emerged *Striga* plant at 10 WAP; SDR8: *Striga* damage ratings at 8 WAP; SDR10: *Striga* damage ratings at 10 WAP

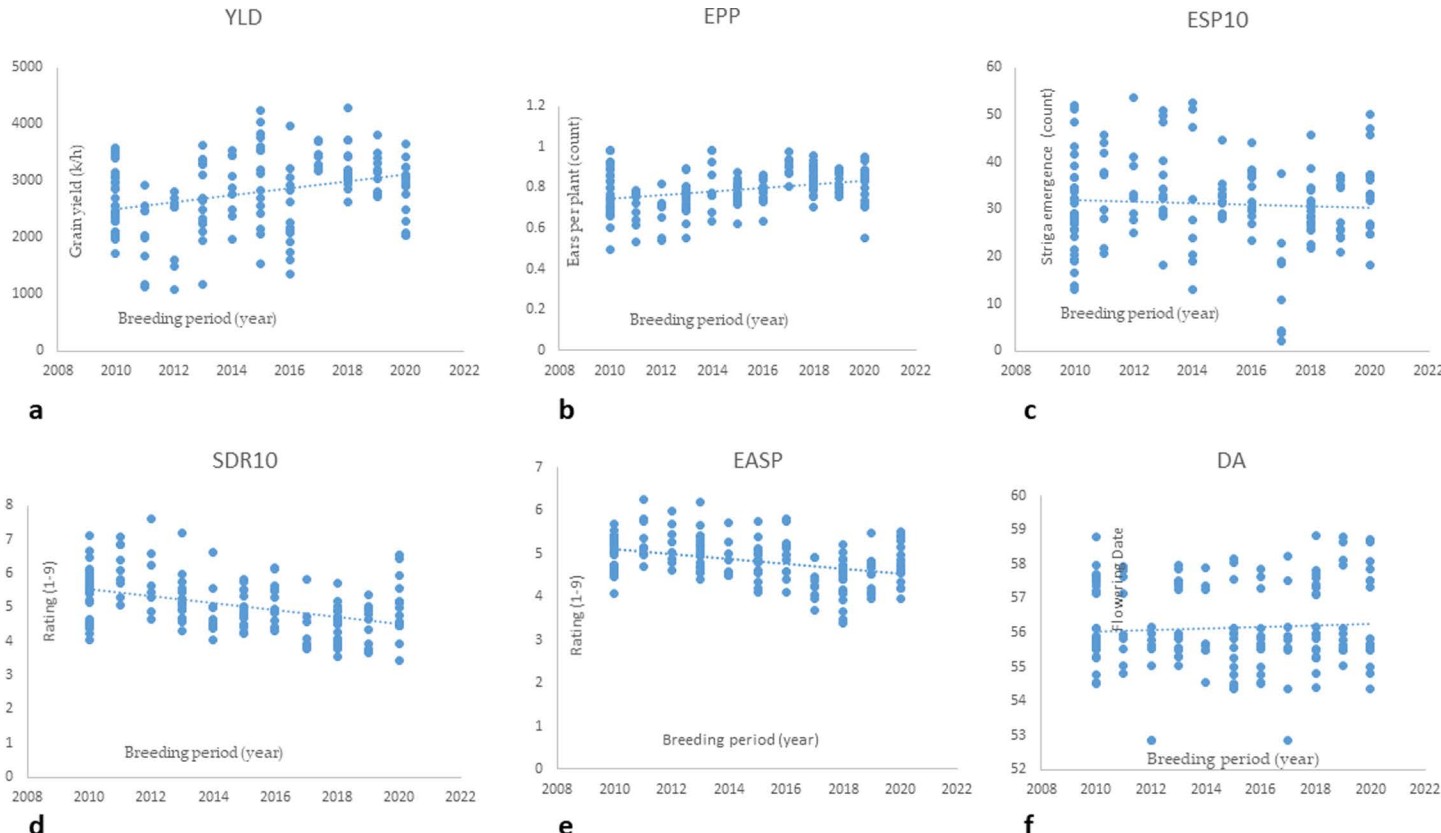

**Fig 1. (a-f) Regression line of grain yield and other Striga adaptive traits (dependent variable) on year of development (independent variable) of hybrids evaluated in regional trials under STRINF conditions at Abuja and Mokwa, 2010–2021.**

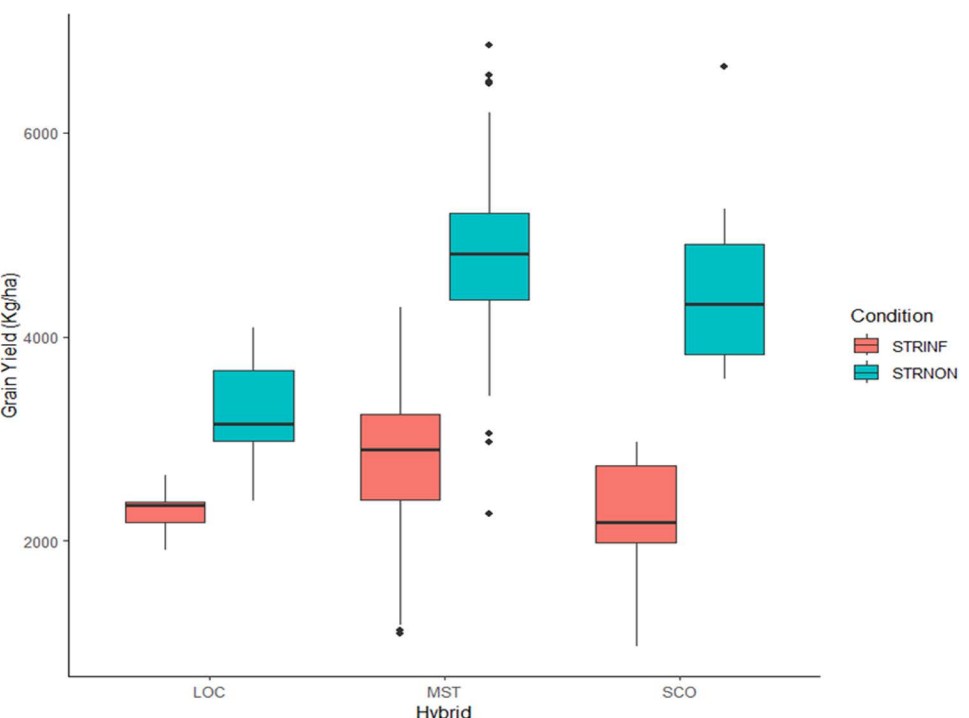

**Fig 2. Box plot of the average grain yield of the MST, SCO, and LOC early maturing maize hybrids evaluated in the regional trials under STRINF and STRNON conditions at Abuja and Mokwa, 2010–2021.**

hybrids had higher mean grain yields than the SCO and the LOC hybrids under STRINF and STRNON conditions, (Fig 2).

## Repeatability estimates of the STRINF and STRNON trials

The results of the repeatability estimates of the trials conducted in the STRINF and STRNON conditions from 2010 to 2021 are shown in Table 3 and Fig 3a−f. Grain yield had repeatability estimates ranging from 0.00 for a trial conducted under STRINF condition at Mokwa in 2016 to 0.91 for the STRINF trial evaluated at Abuja during the 2017 growing season. Among the trials evaluated in Abuja and Mokwa under STRINF and STRNON conditions, only three trials (about 8%) had repeatability estimates of less than 30% for grain yield, about 92% of the trials had repeatability greater than 30% for grain yield, indicating that repeatability estimates for grain yield have moderate to high levels of consistency (Fig 3). Similarly, days to anthesis had repeatability estimates varying from 0.34 at Mokwa under STRINF condition in 2010 to 0.97 at Abuja in 2017 under STRINF condition. Days to silking had had repeatability estimates varied from 0.59 under STRINF condition at Mokwa in 2010 to 0.95 at Abuja in 2017 under STRINF condition and recorded the highest conisistency among the traits recorded. Anthesis silking interval had repeatability ranging from 0.00 at Mokwa under STRNON condition in 2020 to 0.81 at Abuja under STRINF condition in 2016 while repeatability estimates for ear aspect varied from 0.00 at Mokwa under STRNON condition in 2015 to 0.92 at Mokwa under STRINF condition in 2020 and ears per plant had repeatability varying from 0.00 at Mokwa under STRNON condition in 2015 to 0.84 at Abuja under STRINF condition in 2019 (Table 3, Fig 3). Furthermore, repeatability estimates under SRTINF environments varied from 0.10 in 2010 to 0.82 in 2020 at Abuja for ESP8 and from 0.00 in 2015 to 0.82 in 2020 at Abuja for ESP10 and

**Table 3. Repeatability estimate of grain yield and other *Striga*-adaptive traits evaluated under STRINF and STRNON conditions at Mokwa and Abuja, 2010–2021.**

| Trait | Repeatability | | Year | Location | Management |
|---|---|---|---|---|---|
| YLD | minimum | 0.00 | 2016 | Mokwa | STRINF |
| | maximum | 0.91 | 2017 | Abuja | STRINF |
| DA | minimum | 0.34 | 2010 | Mokwa | STRINF |
| | maximum | 0.97 | 2017 | Abuja | STRINF |
| DS | minimum | 0.59 | 2010 | Mokwa | STRINF |
| | maximum | 0.95 | 2017 | Abuja | STRINF |
| ASI | minimum | 0.00 | 2020 | Mokwa | STRNON |
| | maximum | 0.81 | 2016 | Abuja | STRINF |
| EASP | minimum | 0.00 | 2015 | Mokwa | STRNON |
| | maximum | 0.92 | 2020 | Mokwa | STRINF |
| EPP | minimum | 0.00 | 2015 | Mokwa | STRNON |
| | maximum | 0.84 | 2019 | Abuja | STRINF |
| ESP8 | minimum | 0.10 | 2010 | Abuja | STRINF |
| | maximum | 0.82 | 2020 | Abuja | STRINF |
| ESP10 | minimum | 0.00 | 2015 | Abuja | STRINF |
| | maximum | 0.82 | 2020 | Abuja | STRINF |
| SDR8 | minimum | 0.40 | 2010 | Abuja | STRINF |
| | maximum | 0.93 | 2020 | Mokwa | STRINF |
| SDR10 | minimum | 0.52 | 2016 | Mokwa | STRINF |
| | maximum | 0.95 | 2020 | Mokwa | STRINF |

YLD: grain yield; DA: days to 50% anthesis; DS: days to 50% silking; ASI: anthesis-silking interval; EPP: ears per plant; EASP: ear aspect; ESP8: emerged *Striga* plant at 8 WAP; ESP10: emerged *Striga* plant at 10 WAP; SDR8: *Striga* damage ratings at 8 WAP; SDR10: *Striga* damage ratings at 10 WAP

recorded the lowest consistency (Fig 3c) while 0.40 in 2010 at Abuja to 0.93 in 2020 at Mokwa for SDR8 and from 0.52 in 2016 to 0.95 at Mokwa in 2020 for SDR10 (Table 3, Fig 3d).

## Discussion

In addressing the challenges of chronic food insecurity, climate change, environmental sustainability, and improving the economic well-being of farmers in SSA, we used 12 years of historical data to explore three aspects: i) Accelerated genetic gains in grain yield and *Striga* resistance indicator traits of early maturing tropical maize hybrids to explore the most effective use of the genetic variability of early maturing tropical maize populations of IITA. ii) Compare the *performance* of *Striga*-tolerant hybrids developed in the breeding program of IITA to marketed farmer-preferred cultivars and hybrids developed by multinational seed companies and released in many African countries in an effort to justify the program's resource allocation. iii) Assess the repeatability of the trials in the exploration of the genotypic potential of the hybrids under STRINF and STRNON conditions in an effort to develop maize hybrids that could withstand the devastating effects of *Striga* infestation in *Striga*-prone environments.

### Realized genetic gains in grain yield and othe *Striga* adaptive traits

Towards the efforts to make appropriate adjustments in the IITA-MIP, there is need for the investigation of the genotypic potential of the program and introgression of genetic variability into the breeding populations, as well as accelerate the rate of genetic advancement of the breeding program. The results of the present study revealed that significant progress

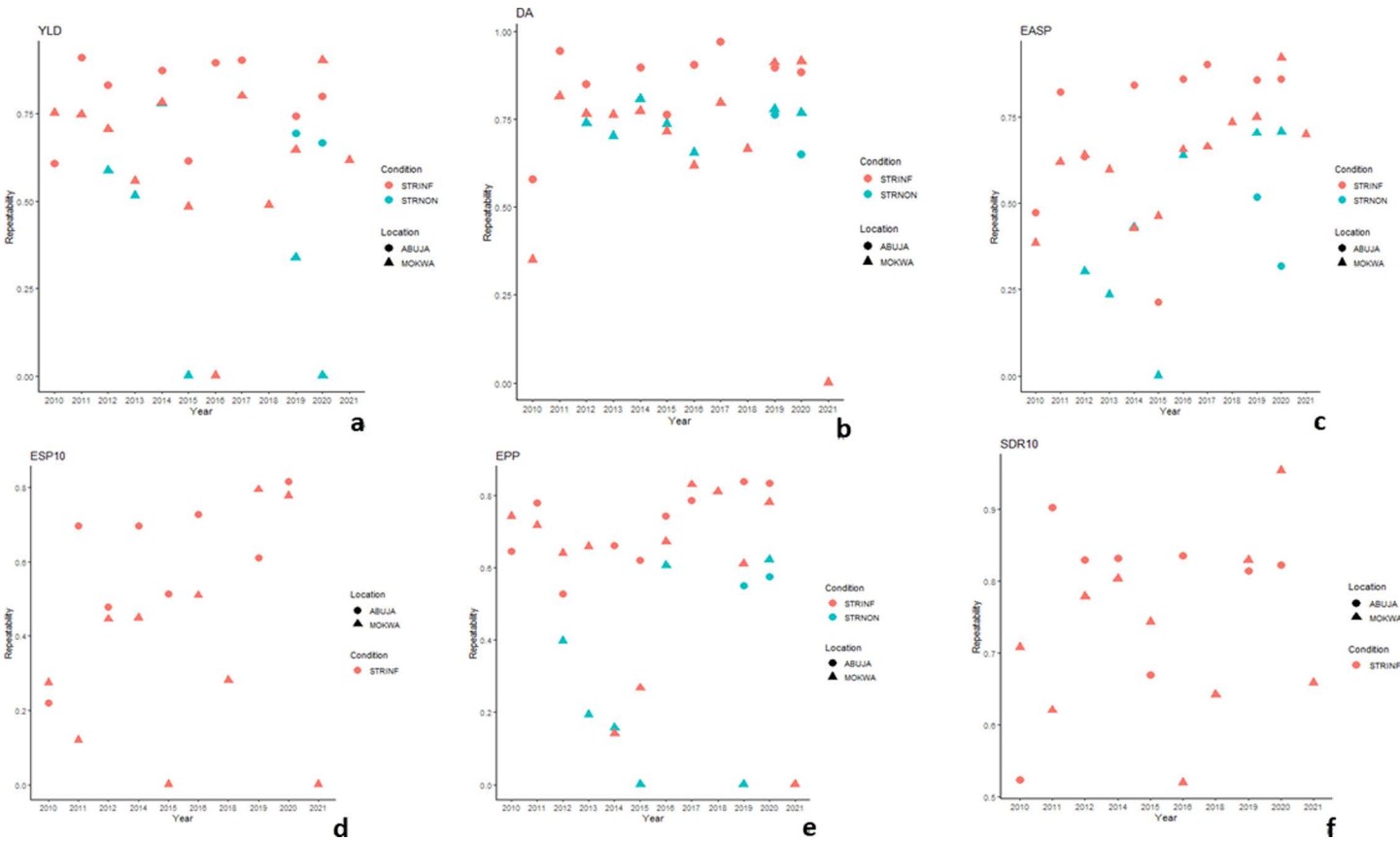

**Fig 3. (a – f) Scattered plot of the repeatability estimates for grain yield and other Striga adaptive traits evaluated under STRINF and STRNON conditions at Abuja and Mokwa 2010 – 2021.**

had been made during the past 12 years of genetic enhancements with annual gains in grain yield reaching 3.40% yr⁻1 and 3.71% yr⁻1 under STRINF and STRNON conditions, respectively. The gains were associated with relative changes in grain yield of 76.87 kg ha⁻1 yr⁻1 and 127.02 kg ha⁻1 yr⁻1 under STRINF and STRNON conditions, respectively. The progress attained in the present study was considerably higher compared to the annual genetic gains in grain yield of 2.91% yr⁻1 and 1.33% yr⁻1 archieved under low and high N environments, respectively, [9], 1.93% yr⁻1 under drought and well-watered conditions [14], and 0.74% yr⁻1 and 1.47% yr⁻1 under drought and full irrigation, respectively [18]. Additionally, Tarekegne *et al.* [27] assessed the genetic gains in grain yield and other traits of the early maturing maize hybrids developed during the period 2000–2018 by International Maize and Wheat Improvement Center (CIMMYT) Southern African breeding program and reported annual yield gains of 2.21%, 2.13%, 1.64% and 1.78%, under low N, managed drought, random stress and optimum conditions. However, Badu-Apraku et al., [1] reported genetic gains of 4.82% yr⁻1 and 1.24% yr⁻1 under STRINF and STRNON conditions, respectively in early maturing maize hybrids, while de la Vega *et al.* [19] reported genetic gains of 2.58% yr⁻1 and 2.30% yr⁻1 under STRINF and STRNON conditions, respectively, in late-maturing maize hybrids. This implied that under *Striga* infestation, the hybrids responded to selection more favorably than under drought or low N conditions. This could be attributed to the fact that under artificial *Striga* infested condition the trials were conducted under more controlled environments compared to drought and low N stress conditions, allowing for better accuracy in the selection

process. Additionally, the *Striga* resistance may be more oligogenic than polygenic which is more commonly observed under drought and low N tolerance. Since the artificial *Striga* infestation process utilized for the development of inbred lines and the evaluation of hybrids has remained unchanged for the past two decades, it could have continuously resulted in high levels of precision, which might have major impact on the achievement of a higher rate of genetic gain in grain yield under STRINF. Alternatively, it could be because more traits were studied during the *Striga* infestation trials than during the drought and low N trials, which could have aided breeders in precisely selecting the most promising hybrids. Furthermore, the low yield improvements observed under drought or low N stresses may be related to the adoption of *Striga*-resistant populations enhanced through S1 recurrent selection without the infusion of tropical donor lines possessing superior genes for drought or low N tolerance as parents of the MST hybrids of IITA-MIP [19]. The significant increases in grain yield were associated with significant increases of 1.84% yr⁻1 and 1.79% yr⁻1 in ears per plant and significant decreases of -1.31% yr⁻1 and -1.75% yr⁻1 in the ear aspect under STRINF and STRNON conditions, respectively. Additionally, the increases in grain yield under STRINF conditions were accompanied by a significant decrease of -3.04% yr⁻1, -2.25% yr⁻1, and -1.27% yr⁻1 in *Striga* damage ratings at 8 and 10 WAP and the number of emerged *Striga* plants at 10 WAP, respectively. Similar results were obtained by Badu-Apraku *et al*. [15] and Menkir *et al*. [18]. Interestingly, the 12 years breeding period did not have effects on the flowering period of the early-maturing maize hybrids.

## Comparative performance of the hybrids

The success of a breeding program depends on the establishment of the right selection criteria and the capacity to select hybrids superior in performance and/or competitive in grain yields under stressful and non-stressful conditions and across diverse environments. The comparative performance of MST hybrids to LOC and SCO hybrids across both STRINF and STRNON conditions revealed a clear distinction between the hybrids of the three groups. The MST hybrids performed exceptionally well in both STRINF and STRNON conditions. The result confirmed that MST with improved levels of *Striga* resistance outperformed those of SCO and LOC hybrids in contrasting environments. Our findings suggested that the breeding strategies, including recurrent selection, backcrossing, hybridization, evaluation, and selection, employed to develop the MST hybrids during the 12-year breeding period were highly effective. This implied that MST hybrids possessed favorable genes that made them display better performers than the SCO and LOC hybrids. Therefore, the accumulation of new favorable alleles through rapid breeding cycles was one of the possible scenarios that significantly increases the rate of genetic gains.

## Assesment of repeatability

Evaluating the repeatability of the STRINF and STRNON trials for assessing and exploring the genotypic potential of the hybrids was the focus of the present study. Unreliable or inconsistent repeatability estimates of the trials could have hindered the progress of accelerated genetic gains, as researchers could have wasted precious time and resources building on flawed or irreproducible findings. Reliable methodologies and experimental designs might have contributed to the overall quality of the scientific experiments. Rigorous methods help ensure that the results are a true reflection of the phenomenon under investigation, rather than being influenced by confounding variables or errors. In a nutshell, repeatability was essential in the assessment of the trials because it ensured the accuracy, consistency, trustworthiness, and validity of the trials. When the random error was restricted, the variations observed in the

trial resulted solely in significant genotypic differences, environment, and genotype × environment (G×E) interactions. Nevertheless, the overall observed variation in the experiment was considerably influenced by genotype, environment, and G×E interactions, determining the repeatability of the trial, considering the combined effects of genotype, environment, G×E interactions, and random errors. Therefore, the disparities in the repeatability estimates of the trials conducted over a long period may be mostly due to the experimental procedures used/ random error. We estimated the repeatability of the STRINF and STRNON trials conducted between 2010 and 2021 for each of the measured traits.

The highest repeatability was observed for ears per plant and flowering dates ([Fig 3]), indicating that these traits would not be influenced largely by the environment and the G×E interactions. This suggested that flowering dates and ears per plant could be used to evaluate the effectiveness of the STRINF and STRNON trials in the exploration of the genotypic potential of the hybrids. However, the lowest average repeatability was observed with the emerged *Striga* plant at 8 and 10 WAP. Thus, the emerged *Striga* plant at 8 and 10 WAP should not be used to determine the efficiency of the STRINF and STRNON trials in the exploration of the genotypic potential of the hybrids. This is not surprising because the maize hybrids had different levels of stimulations by the strigolactones, which have led to uneven germination of *S*triga seeds and their attachments to the host [28]. Additionally, a combination of factors encountered during trial evaluation at the different locations including distinct soil properties, nutrient contents, and pH levels, could have affected the emergence of the *Striga* plant [29]. Generally, a good level of repeatability was observed in the early maturing maize trials conducted under STRINF and STRNON conditions conducted between the 2010 and 2021 growing seasons, indicating the potential of the trials in the identification of promising hybrids for commercialization in the SSA for increased food security and poverty alleviation.

## Conclusions

The present study used 12 years of historical data to analyse the genetic gains in grain yield and associated changes in yield-related traits under STRINF and STRNON conditions and to estimate the repeatability of the of the studied traits. Annual genetic gains in grain yield of 3.40% yr$^{-1}$ and 3.71% yr$^{-1}$ were obtained under STRINF and STRNON conditions, respectively. The flowering dates and ears per plant had the highest repeatability estimates. These traits have the lowest multi-dimensionalities, independent of environment influence and as such suitable for assessing repeatability of the trials. The emerged *Striga* plant at 8 and 10 WAP were identified as unsuitable traits for determining the efficiency of the trials particularly in exploring the genotypic potential of the hybrids. Overall, significant genetic gains were achieved, and a good level of repeatability was observed in the early maturing maize trials. These findings confirm that the quality of the STRINF and STRNON trials was sufficient for identifying outstanding hybrids suitable for commercialization in SSA to mitigate food insecurity and alleviate poverty in the region.

## Supporting information

**S1 Files.  Supplementary Tables S1 and S2.**
(DOCX)

## Acknowledgments

The authors are grateful to the IITA-MIP, EBS, and Biometric Unit technical, field, and administrative staff for providing technical, field, and administrative assistance.

## Author contributions

**Conceptualization:** Adamu Masari Abubakar, Idris Ishola Adejumobi, Baffour Badu-Apraku.

**Data curation:** Adamu Masari Abubakar, Idris Ishola Adejumobi, Kayode Rapheal Fowobaje, Simon Imoro.

**Formal analysis:** Kayode Rapheal Fowobaje, Ibnou Dieng.

**Funding acquisition:** Baffour Badu-Apraku.

**Methodology:** Adamu Masari Abubakar, Simon Imoro.

**Supervision:** Ibnou Dieng, Baffour Badu-Apraku.

**Writing – original draft:** Adamu Masari Abubakar, Zakariya Saminu.

**Writing – review & editing:** Adamu Masari Abubakar, Idris Ishola Adejumobi, Kayode Rapheal Fowobaje, Ibnou Dieng, Zakariya Saminu, Simon Imoro, Baffour Badu-Apraku.

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
