## [Decision Letter · Decision Letter 0]

28 Nov 2024

PONE-D-24-47860REPEATABILITY AND GENETIC ADVANCEMENT IN EARLY MATURING MAIZE HYBRID TRIALS CONDUCTED UNDER STRIGA-INFESTED AND NON-INFESTED CONDITIONSPLOS ONE

Dear Dr. Adejumobi,

Thank you for submitting your manuscript to PLOS ONE. After careful consideration, we feel that it has merit but does not fully meet PLOS ONE’s publication criteria as it currently stands. Therefore, we invite you to submit a revised version of the manuscript that addresses the points raised during the review process. Please submit your revised manuscript by Jan 12 2025 11:59PM. If you will need more time than this to complete your revisions, please reply to this message or contact the journal office at plosone@plos.org . Please include the following items when submitting your revised manuscript:

We look forward to receiving your revised manuscript.

Kind regards,

C Anilkumar, Ph.D.

Academic Editor

PLOS ONE

Journal Requirements:

4. Thank you for stating the following financial disclosure: “This research was supported by the Bill & Melinda Gates Foundation under the Accelerated Genetic Gains (AGG) and Stress Tolerant Maize for Africa (STMA) Projects.”

5. Thank you for uploading your study's underlying data set. Unfortunately, the repository you have noted in your Data Availability statement does not qualify as an acceptable data repository according to PLOS's standards. At this time, please upload the minimal data set necessary to replicate your study's findings to a stable, public repository (such as figshare or Dryad) and provide us with the relevant URLs, DOIs, or accession numbers that may be used to access these data. For a list of recommended repositories and additional information on PLOS standards for data deposition, please see https://journals.plos.org/plosone/s/recommended-repositories .

7. We note you have included a table to which you do not refer in the text of your manuscript. Please ensure that you refer to Table 1 in your text; if accepted, production will need this reference to link the reader to the Table.

8. Please include captions for your Supporting Information files at the end of your manuscript, and update any in-text citations to match accordingly. Please see our Supporting Information guidelines for more information: "http://journals.plos.org/plosone/s/supporting-information.

Additional Editor Comments:

Dear authors,

The manuscript was reviewed by two subject specialists and myself. The manuscript needs a minor revision on the technical content of the manuscript, and it needs to be edited thoroughly for English language. Upon revision, I will be happy to accept the manuscript.

Reviewers' comments:

Reviewer's Responses to Questions

**Comments to the Author**

1. Is the manuscript technically sound, and do the data support the conclusions?

Reviewer #1: Yes

Reviewer #2: Yes

2. Has the statistical analysis been performed appropriately and rigorously? 

Reviewer #1: Yes

Reviewer #2: Yes

3. Have the authors made all data underlying the findings in their manuscript fully available?

Reviewer #1: Yes

Reviewer #2: No

4. Is the manuscript presented in an intelligible fashion and written in standard English?

Reviewer #1: Yes

Reviewer #2: Yes

5. Review Comments to the Author

Reviewer #1: The quality of the manuscript is technically sound, and data addressed all objectives. However, some concerns are:

1. Results should be reported in the past tense.

2. Some minor grammatical errors need correction; a few have already been pointed out.

3. Figures addressing Objective 2 are missing.

4. Please refer to Table 3 and recast your results on repeatability values for the traits.

5. Indicate in the Materials and Methods section the method used for assessing the significance of values.

6. Title can be rephrased as "Repeatability and genetic gain"

7. Restructure your conclusions to address the objectives.

Reviewer #2: The manuscript is well written except for few MINOR REVISIONS to be made before acceptance:

(1) The 'Introduction' section is too long and need to be shortened with good flow of the required information

(2) Correct as 'Materials and Methods'

(3) Present the coordinates of the test locations

(4) Present the agro-climatic pattern of the different locations and years

(5) The discussion to be presented with suitable sub-headings for easy understanding to the readers

6. PLOS authors have the option to publish the peer review history of their article (what does this mean? ). If published, this will include your full peer review and any attached files.

**Do you want your identity to be public for this peer review?** For information about this choice, including consent withdrawal, please see our Privacy Policy .

Reviewer #1: No

Reviewer #2: No

---

## [Author Response · Author response to Decision Letter 1]

6 Dec 2024

REBUTTAL TO REVIEWERS’ COMMENTS ON THE MANUSCRIPT TITLED “ REPEATABILITY AND GENETIC ADVANCES IN EARLY MATURING MAIZE HYBRID TRIALS CONDUCTED UNDER STRIGA-INFESTED AND NON-INFESTED CONDITIONS”

Response to Editor (Journal requirements)

Response: The manuscript has been modified to meet PLOS ONE's style requirements.

Response: The funding-related text has been removed from the manuscript

Response: The funding information will be corrected upon resubmission of the revised manuscript.

4. Thank you for stating the following financial disclosure: “This research was supported by the Bill & Melinda Gates Foundation under the Accelerated Genetic Gains (AGG) and Stress Tolerant Maize for Africa (STMA) Projects.”

Response: The funder was the main donor of the AGG and STMA projects and also the payer of the article processing charge for the manuscript.

5. Thank you for uploading your study's underlying data set. Unfortunately, the repository you have noted in your Data Availability statement does not qualify as an acceptable data repository according to PLOS's standards. At this time, please upload the minimal data set necessary to replicate your study's findings to a stable, public repository (such as figshare or Dryad) and provide us with the relevant URLs, DOIs, or accession numbers that may be used to access these data. For a list of recommended repositories and additional information on PLOS standards for data deposition, please see https://journals.plos.org/plosone/s/recommended-repositories

Response: The acceptable data repository according to PLOS's standards with DOIs have now been provided in the data availability statement of the manuscript

Response: The data have now been made available.

7. We note you have included a table to which you do not refer in the text of your manuscript. Please ensure that you refer to Table 1 in your text; if accepted, production will need this reference to link the reader to the Table.

Response: We have made appropriate reference to Table 1 in the revised version of the manuscript

Response: Captions for Supporting Information files have now been provided at the end of the manuscript

Response: The list of references has been reviewed to ensure that it is complete and correct. We observed that a reference was not cited appropriately which was taken care of. In addition, duplicate references have also been resolved in the revised version of the manuscript.

Responses to Reviewers' Comments

Reviewer #1: The quality of the manuscript is technically sound, and data addressed all objectives. However, some concerns are:

1. Results should be reported in the past tense.

Response: The results section of the manuscript has been reported in the past tense as requested.

2. Some minor grammatical errors need correction; a few have already been pointed out.

Response: We have revised the manuscript for grammatical modifications and sentence chronology to improve readability in the revised version of the manuscript as requested.

3. Figures addressing Objective 2 are missing.

Response: We apologize for this however all figures relevant to the work were submitted separately to the PLOS ONE webpage. We hope the editor can resolve this issue in the subsequent interactive review.

4. Please refer to Table 3 and recast your results on repeatability values for the traits.

Response: As requested, we have recast the results relating to Table 3 to ensure accurate interpretation for readers.

5. Indicate in the Materials and Methods section the method used for assessing the significance of values.

Response: The method used for assessing the significance of the values have been included in the Materials and Methods section as requested in the revised manuscript.

6. Title can be rephrased as "Repeatability and genetic gain"

Response: We appreciate your suggestion to improve the quality of the manuscript. The co-authors have several similar publications with titles including the word “genetic gains”. We would like to use the word “genetic advances” as a replacement for “genetic advancement” to provide a sense of uniqueness to the manuscript title.

7. Restructure your conclusions to address the objectives.

Response: We have restructured the conclusion section to focus on the objectives in the revised manuscript as requested.

Reviewer #2: The manuscript is well written except for few MINOR REVISIONS to be made before acceptance:

(1) The 'Introduction' section is too long and need to be shortened with good flow of the required information

Response: Thank you. We have revised the Introduction as requested.

(2) Correct as 'Materials and Methods'

Response: Material and Method were corrected as 'Materials and Methods' as requested.

(3) Present the coordinates of the test locations

Response: Coordinates of the test locations have been referenced in the revised manuscript as requested.

(4) Present the agro-climatic pattern of the different locations and years

Response: Thank you for your comment. The agro-climatic patterns of the two locations have been included as a supplementary file to be submitted alongside the revised manuscript.

(5) The discussion to be presented with suitable sub-headings for easy understanding to the readers

Response: Discussion has now been presented with suitable sub-headings as requested.

---

## [Decision Letter · Decision Letter 1]

23 Dec 2024

PONE-D-24-47860R1REPEATABILITY AND GENETIC ADVANCEMENT IN EARLY MATURING MAIZE HYBRID TRIALS CONDUCTED UNDER STRIGA-INFESTED AND NON-INFESTED CONDITIONSPLOS ONE

Dear Dr. Adejumobi,

Thank you for submitting your manuscript to PLOS ONE. After careful consideration, we feel that it has merit but does not fully meet PLOS ONE’s publication criteria as it currently stands. Therefore, we invite you to submit a revised version of the manuscript that addresses the points raised during the review process.

**ACADEMIC EDITOR: ** The manuscript is updated in the revision. However, one of our reviewers suggested minor corrections in the abstract and results sections. It is advised to revise the manuscript following reviewer suggestions.

We look forward to receiving your revised manuscript.

Kind regards,

C Anilkumar, Ph.D.

Academic Editor

PLOS ONE

Journal Requirements:

Additional Editor Comments:

The manuscript is updated in the revision. However, one of our reviewers suggested minor corrections in the abstract and results sections. It is advised to revise the manuscript following reviewer suggestions.

Reviewers' comments:

Reviewer's Responses to Questions

**Comments to the Author**

1. If the authors have adequately addressed your comments raised in a previous round of review and you feel that this manuscript is now acceptable for publication, you may indicate that here to bypass the “Comments to the Author” section, enter your conflict of interest statement in the “Confidential to Editor” section, and submit your "Accept" recommendation.

Reviewer #1: All comments have been addressed

Reviewer #2: All comments have been addressed

2. Is the manuscript technically sound, and do the data support the conclusions?

Reviewer #1: Yes

Reviewer #2: Yes

3. Has the statistical analysis been performed appropriately and rigorously? 

Reviewer #1: Yes

Reviewer #2: Yes

4. Have the authors made all data underlying the findings in their manuscript fully available?

Reviewer #1: Yes

Reviewer #2: Yes

5. Is the manuscript presented in an intelligible fashion and written in standard English?

Reviewer #1: Yes

Reviewer #2: Yes

6. Review Comments to the Author

Reviewer #1: 1. Please refer to Table 3 and recast your result in abstract and result section. "The results indicated that the number of emerged Striga plants at 8 and 10 WAP recorded the lowest repeatability". What of ASI with 0.81?

2. Table 2, correct spelling of Genetic gain (yearr) to "year"

3. Annual genetic gains of 3.40% yr−1 and 3.71 % yr−1 were obtained under STRINF and STRNON conditions,

respectively"for what trait??????". Please indicate clearly.

4. Recast the last paragraph in conclusion as indicated in the attached manuscript.

Reviewer #2: The authors have addressed the comments suggested earlier and have substantially improved the manuscript.

7. PLOS authors have the option to publish the peer review history of their article (what does this mean? ). If published, this will include your full peer review and any attached files.

**Do you want your identity to be public for this peer review?** For information about this choice, including consent withdrawal, please see our Privacy Policy .

Reviewer #1: No

Reviewer #2: No

---

## [Author Response · Author response to Decision Letter 2]

28 Jan 2025

REBUTTAL TO REVIEWERS’ COMMENTS ON THE MANUSCRIPT TITLED “REPEATABILITY AND GENETIC ADVANCES IN EARLY MATURING MAIZE HYBRID TRIALS CONDUCTED UNDER STRIGA-INFESTED AND NON-INFESTED CONDITIONS”

Response to Reviewers' Comments

Reviewer #1

Comment 1

Please refer to Table 3 and recast your result in abstract and result section. "The results indicated that the number of emerged Striga plants at 8 and 10 WAP recorded the lowest repeatability". What of ASI with 0.81?

Response 1

Thank you for your suggestion to improve the quality of this manuscript. The result in Table 3 presented the ranges of the repeatability estimates. Still, Fig 3 indicated the consistency in the repeatability estimates and therefore, the lowest repeatability here referred in terms of consistency and that has now been addressed clearly.

Comment 2

Table 2, correct spelling of Genetic gain (yearr) to "year"

Response 2

This has been effected in the revised version of the manuscript. Thanks

Comment 3

Annual genetic gains of 3.40% yr−1 and 3.71 % yr−1 were obtained under STRINF and STRNON conditions, respectively"for what trait??????". Please indicate clearly.

Response 3

The traits have been included in the sentence as requested. Thanks.

Comment 4

Recast the last paragraph in conclusion as indicated in the attached manuscript.

Response 4

Modified as suggested. We appreciate your observation.

---

## [Decision Letter · Decision Letter 2]

31 Jan 2025

REPEATABILITY AND GENETIC ADVANCEMENT IN EARLY MATURING MAIZE HYBRID TRIALS CONDUCTED UNDER STRIGA-INFESTED AND NON-INFESTED CONDITIONS

PONE-D-24-47860R2

Dear Dr. Adejumobi,

We’re pleased to inform you that your manuscript has been judged scientifically suitable for publication and will be formally accepted for publication once it meets all outstanding technical requirements.

Kind regards,

Anilkumar, Ph.D.

Academic Editor

PLOS ONE

Reviewers' comments:

Reviewer's Responses to Questions

**Comments to the Author**

1. If the authors have adequately addressed your comments raised in a previous round of review and you feel that this manuscript is now acceptable for publication, you may indicate that here to bypass the “Comments to the Author” section, enter your conflict of interest statement in the “Confidential to Editor” section, and submit your "Accept" recommendation.

Reviewer #1: All comments have been addressed

2. Is the manuscript technically sound, and do the data support the conclusions?

Reviewer #1: Yes

3. Has the statistical analysis been performed appropriately and rigorously? 

Reviewer #1: Yes

4. Have the authors made all data underlying the findings in their manuscript fully available?

Reviewer #1: Yes

5. Is the manuscript presented in an intelligible fashion and written in standard English?

Reviewer #1: Yes

6. Review Comments to the Author

Reviewer #1: (No Response)

7. PLOS authors have the option to publish the peer review history of their article (what does this mean? ). If published, this will include your full peer review and any attached files.

**Do you want your identity to be public for this peer review?** For information about this choice, including consent withdrawal, please see our Privacy Policy .

Reviewer #1: No

---

## [Editor Report · Acceptance letter]

PONE-D-24-47860R2

PLOS ONE

Dear Dr. Adejumobi,

I'm pleased to inform you that your manuscript has been deemed suitable for publication in PLOS ONE. Congratulations! Your manuscript is now being handed over to our production team.

Kind regards,

on behalf of

Dr. C Anilkumar

Academic Editor

PLOS ONE